# Accuracy of Manual Snow Sampling, Depending on the Sampler's Cross-Section—A Comparative Study

Marko Kaasik [1,*], Outi Meinander [2], Leena Leppänen [2,3], Kati Anttila [4], Pavla Dagsson-Waldhauserova [5,6], Anders Ginnerup [7], Timo Hampinen [8], Yijing Liu [9], Andri Gunnarsson [10], Kirsty Langley [7] and Ali Nadir Arslan [2]

1   Institute of Physics, University of Tartu, 50411 Tartu, Estonia
2   Finnish Meteorological Institute, 00560 Helsinki, Finland; outi.meinander@fmi.fi (O.M.);
    leena.leppanen@ulapland.fi (L.L.); ali.nadir.arslan@fmi.fi (A.N.A.)
3   Arctic Centre, University of Lapland, 96100 Rovaniemi, Finland
4   Finnish Environment Institute, 00790 Helsinki, Finland; kati.anttila@syke.fi
5   Faculty of Environmental and Forest Sciences, Agricultural University of Iceland,
    311 Borgarnes, Iceland; pavla@lbhi.is
6   Faculty of Environmental and Forest Sciences, Czech University of Life Sciences Prague,
    16500 Prague, Czech Republic
7   Asiaq—Greenland Survey, Qatserisut 8, Nuuk 3900, Greenland; ang@asiaq.gl (A.G.); kal@asiaq.gl (K.L.)
8   North Ostrobothnia Centre for Economic Development, Transport, and the Environment, Veteraanikatu 1,
    9013 Oulu, Finland; timo.hampinen@ely-keskus.fi
9   Department of Geosciences and Natural Resource Management, University of Copenhagen,
    1350 Copenhagen, Denmark; yili@ign.ku.dk
10  Landsvirkjun, Háaleitisbraut 68, 103 Reykjavík, Iceland; andrigun@lv.is
*   Correspondence: marko.kaasik@ut.ee; Tel.: +372-5206174

**Abstract:** Snow sampling, either by inserting a tube through the entire snowpack or by taking samples from the vertical profile, is widely applied to measure the snow depth, density, and snow water equivalent (SWE). A comparative study of snow-sampling methods was carried out on 24 March 2022 in Sodankylä, Finland. Six groups from five countries (Estonia, Finland, Greenland, Iceland, and Sweden) participated, using 12 different snow samplers, including 9 bulk tube samplers and 3 density cutters. The cross-sectional area of the SWE samplers varied from 11 to 100 cm$^2$, while tube length varied from 30 cm to 100 cm. The cross-sectional area of the density profile cutters varied from 100 cm$^2$ to 200 cm$^2$ and the vertical sampling step varied from 5 cm to 10 cm. The samples were taken from snow pits in 55–65-centimeter-deep snow cover in a flat area with sparse pine trees, with the pits at a maximum distance of 10 m from each other. Each tube sampling series consisted of 3–10 vertical columns to ensure statistical validation. The snowpack was relatively soft, with two moderately hard crust layers. The density recorded in the tube sample measurements varied from 218 to 265 kgm$^{-3}$. The measurement results of SWE, however, varied depending on the sampling equipment used, ranging from 148 to 180 kgm$^{-2}$, with two outliers of 77 and 106 kgm$^{-2}$, both with 11 cm$^2$ samplers.

**Keywords:** snow cover; snow sampling; snow water equivalent; tube sampler

## 1. Introduction

Boreal forest, comprising a normally snow-covered area in winter, extends over 14 million km$^2$, constituting 8% of the global land area [1]. European boreal and temperate forests, growing on 1.6 million km$^2$ of Europe (excluding Russia and Turkey) are expected to be subject to large disturbances due to climate warming [2]. Snow conditions at the bases of boreal forests are important from several aspects. It has been found that the albedo of a boreal forest without snow is about 0.11, but it increases threefold (up to 0.33–0.35) if snow cover is present [3].

Despite the prevalence of and the widening capabilities offered by remote sensing, in situ methods remain vital to the collecting of data on the physical and chemical properties of snow cover. In situ measurements serve as reference and ground-truth data for satellite-derived snow products, offering accurate descriptions and the assimilation of snow information into hydrological, land surface, meteorological, and climate models and the chemical analysis of impurities [4–7]. When developing the snow water equivalent (SWE) prediction models, it was found that the uncertainties of snow depth are highest in forested areas with a sparse in situ measurement network [8]. In situ measurements remain the main reference point used in the validation of novel snow depth measurement techniques, such as drone-borne LiDAR scanning [9] and more conventional satellite-based methods [10]. Complex SWE modeling techniques include both in situ and remote sensing data, which are used to represent the natural variability due to the geographical extent and the type of landscape [11]. Snow pit measurements serve as a validation measure for the recently developed and less time-consuming hand-held radar measurement techniques of snow stratigraphy (using the vertical snow density profile) [12].

In situ measurements vary considerably, depending on their purpose, the instruments used, and the observers at different institutions [6]. Snow sampling, either by inserting a tube through the entire snowpack or by taking samples layer by layer (or with constant interval), is widely applied to measure the snow depth, density, and SWE. SWE is defined as the depth of water that will, theoretically, result if the whole snow package instantaneously melts. SWE can be expressed either as the depth of the water layer (usually given in millimeters) or as water mass per surface area (e.g., $kgm^{-2}$).

There are few scientific articles comparing snow water-equivalent measurements that have been obtained from different classes of equipment, including SWE tubes and density cutters. Proksch et al. [13] concluded that snow densities measured by different methods, including the different density cutters and micro-CT, agreed to within 9%. A previous comparative study of SWE samplers showed that the devices provided slightly different uncertainties since they are designed for use in different snow conditions [14]. The study showed that the uncertainty induced by instrumental bias was generally less than 10%. Another comparison of SWE, measured with both bulk samplers and density cutters, concluded that the density cutters overestimated SWE [15]. Dixon and Boon [16] also compared different SWE sampling systems, whereby a 250 $cm^2$ density cutter resulted in overestimation. In addition, they showed that the Federal Sampler diverged from two other samplers with larger cross-sectional areas. In general, all studies indicated that the error sources in bulk SWE measurements are typically related to sampler design, observer differences, weighing, and snow conditions, e.g., [16–20].

Snow samples are often used in environmental research to quantify the deposition of impurities from the outdoor air during the accumulation of snow cover. Impurities can change the water-holding capacity of melting snow; therefore, snow density and SWE can be connected to the levels of impurities in the snow. When the concentrations of impurities in snow are analyzed in a snow sample, they are determined for a known amount of snow, such as in [μg/kg], or according to the volume of the melted snow in [μg/L], which, in the latter case, would equal a measurement in parts per billion (ppb), for example [13]. The accuracy of the chemical analysis depends on multiple factors, such as the analysis protocols [21], and is indirectly dependent on the accuracy of snow sampling, as highlighted in this study. However, when the total number of impurities in a snowpack is calculated for the accumulated deposition load, on the basis of the chemical analysis of snow samples, the accuracy when determining the snow depth and other snowpack properties, such as SWE or density [15], may play an indirect role. Meinander et al. [22] hypothesized that impurities can change the water-holding capacity of melting snow and, thus, snow density and that SWE can be connected to the number of impurities found in the snow. The snow impurity concentrations in a snowpack also vary temporally, spatially, and horizontally, according to impurity deposition, snow metamorphosis processes, and meteorological conditions [23].

Herein, a comparative study of snow sampling methods is presented, which includes both bulk-density tube samplers and layer-wise density cutters. The study was carried out on 24 March 2022 under the auspices of the Nordic Snow Network (NordSnowNet) in Sodankylä, Finland. The Nordic Snow Network (https://nordsnownet.fmi.fi/ (accessed on 27 June 2023)) represents a cooperation initiative between the national meteorological institutes and universities in Norway, Denmark, Finland, Sweden, Iceland, Estonia, and Greenland. The network is dedicated to the research of snow in Nordic regions and the related Arctic areas, aiming at enhancing the harmonization of the measurement practices of snow information for researchers, data users, and education communities.

## 2. Materials and Methods

The Arctic Space Centre of the Finnish Meteorological Institute in Sodankylä is host to numerous manual and automated snow measurement systems [24]. The experiment was conducted in the Intensive Observation Area, which is one of the snow measurement sites included at the research station.

Research groups from Denmark (University of Copenhagen), Estonia (University of Tartu), Finland (the Finnish Meteorological Institute, Finnish Environment Institute, Arctic Centre, and North Ostrobothnia ELY-centre), Greenland (Asiaq—Greenland Survey) and Iceland (Agricultural University of Iceland), participated using 12 different snow samplers, including nine bulk tube samplers and three density profile cutters (Figure 1). The cross-section areas of the tube samples varied within a range of 11 to 100 cm². In the case of profile sampling, the vertical sampling step was 5 or 10 cm, with an area of 100 cm² or 200 cm². The snow pits were located at a maximum of 10 m from each other, in a flat area with sparse pine trees (Figure 2). The snow cover was 55–65 cm high. Each tube sampling series consisted of 3–10 vertical columns for statistical validation. The density profile cutters were obtained from a single column. In addition, the stratigraphy of the layers, grain type, wetness, and hardness were observed and recorded.

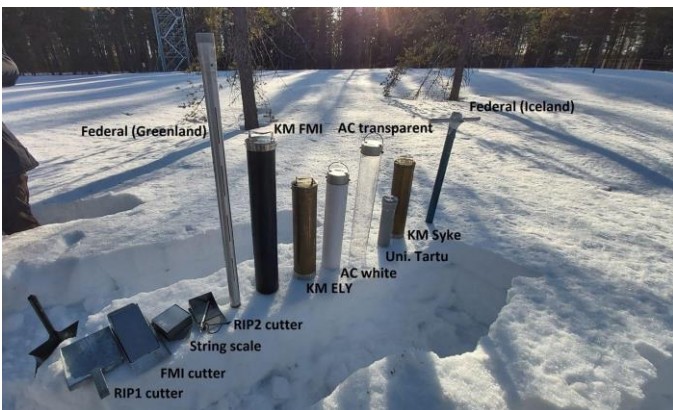

**Figure 1.** The tube samplers and density cutters used in the study.

The Korhonen–Melander snow sampler has been used in Finland since the 1920s for the purposes of SWE measurement. The Korhonen–Melander snow sampler used by the Finnish Meteorological Institute (FMI) was a black plastic tube with a length of 70 cm and a cross-sectional area of 100 cm². The Korhonen–Melander snow samplers used by the Finnish Environment Institute (Syke) and the North Ostrobothnia ELY-centre (Centre for Economic Development, Transport, and the Environment) were metallic tubes with a length of 50 cm and a cross-sectional area of 100 cm². One end of the sampler was sealed with a removable lid. The open end of the sampler had a sharpened metal frame. The sampler was cleaned of snow before the measurement was taken. Then, the sampler was inserted perpendicularly into the snowpack and a sample was collected, using a small shovel to cover the open end. Snow depth was recorded, either according to the scale on the side of the sampler or with a ruler. If the snowpack was deeper than the height of the sampler,

several samples were taken to ensure sampling of the full depth. The sample was weighed in the sampling tube using a mechanical scale calibrated to show the SWE value directly.

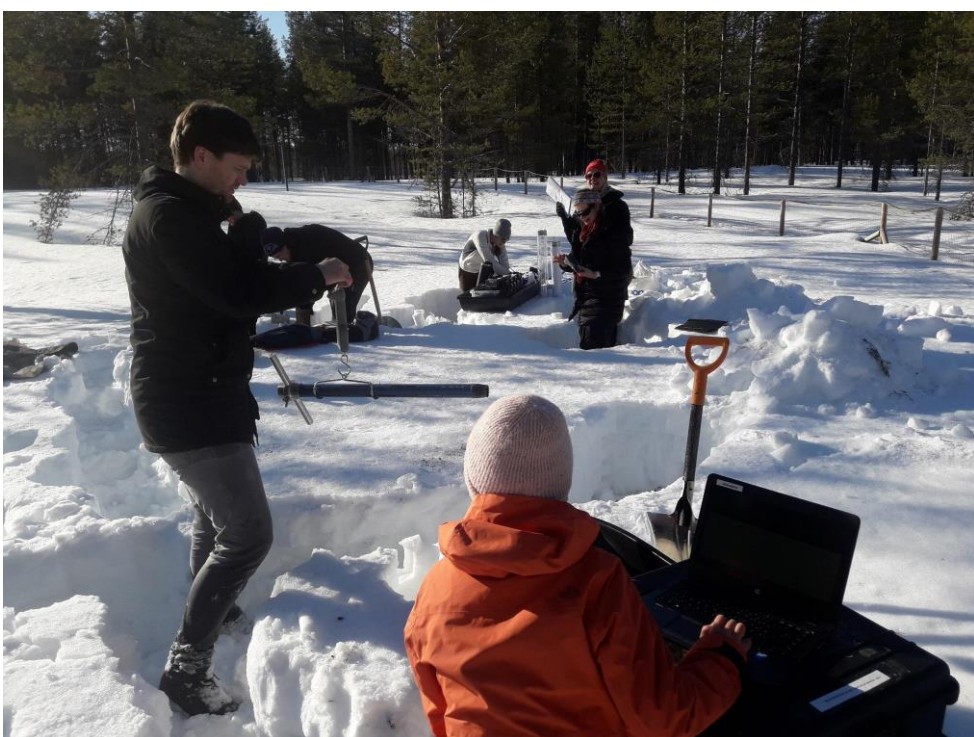

**Figure 2.** A moment captured during snow sampling fieldwork, showing the measurement location in a flat open area of the forest, inside a fence.

The snow samplers used by the Arctic Centre of the University of Lapland were based on the same measurement principle as the Korhonen–Melander snow samplers. The transparent snow sampler (AC transparent) was made of 3-millimeter-thick polycarbonate. The length of the sampler was 65 cm and the cross-sectional area was 69 cm$^2$. The open edge of the tube was slightly sharpened. The white plastic snow sampler (AC white) was slightly shorter (54 cm) and the cross-sectional area was 79 cm$^2$. The open edge of the tube was not sharpened since the thickness of the plastic was only ~1 mm. One end of the samplers was sealed with a removable lid. The snow samples were weighed in a plastic bag, using an electronic scale (Patriot; accuracy of 5 g; up to 50 kg). The weight of an empty bag was also recorded. The sampling procedure was similar to that of the KM samplers, except for the fact that the sample was weighed in a bag and the snow depth was measured using a ruler.

The deposition sampler from the University of Tartu (which was used to take samples for the measurement of the deposition of air pollutants) is made of a standard gray PVC sewage tube (Upnor) that is 7 cm in diameter; thus, it has a cross-sectional area of 38 cm$^2$. One end of the sampler is sealed with a removable lid. The sampler is 28 cm long; thus, each sample in this study consisted of two cores from the same vertical column. The sampling procedure was similar to that used with the Korhonen–Melander-type samplers. The snow sample was weighed in a plastic bag, using the same electronic scale as that used by the Arctic Centre.

The Federal Sampler has a length of 1 m and it is possible to add extensions when the snowpack depth is greater than 1 m. The cross-sectional area of the sampler was only 11 cm$^2$, which was much smaller than those of the other samplers used in this study. The Federal Sampler has cutter teeth to penetrate hard snow and ice layers. The sampler was inserted perpendicularly into the snowpack until the ground was reached, after which the sampler was twisted to collect a soil plug that closed the end of the tube. The sampler could, thus, be lifted from the snowpack without digging, and is weighed with an electronic scale.

Snow depth was recorded using a centimeter scale on the side of the sampler. Two such samplers, from the Greenland Survey and the Agricultural University of Iceland, were used in this study.

Density profiles were measured using three different density cutters. The SWE values were calculated from the measurements that were taken. The first cutter was a 250 cm$^3$ RIP2 cutter (Snowmetrics; see Figure 3). The height of the wedge-shaped cutter was 5 cm. The snow sample was extracted along with a lid. A total of nine measurements were made, typically with an interval of 5 cm from the snow surface to the ground. The samples were weighed in a plastic bag, using a mechanical spring balance (Pesola AG). The RIP2 sampling location presented difficult sampling conditions, due to the vegetation growing in the 0–10 cm snow layer. Since we could not assume that there was no change in the snow density below 10 cm, the following options were available: to not analyze data from the bottom layers, or to give the best estimate for the bottom layer. We chose here not to include the data from the bottom layer with the vegetation. The second cutter was a wedge-shaped RIP1 cutter (Snowmetrics) with a volume of 1000 cm$^3$ and a height of 10 cm. The snow sample was also extracted with a lid. Samples were taken every 10 cm and six samples were collected. The samples were weighed using an electronic scale (A&D HT-3000, readability 1 g). The third density profile was measured using a density cutter made at FMI. The height of the aluminum rectangular cuboid cutter is 5 cm and the volume is 500 cm$^3$. Samples were extracted, using lids to cover both open ends of the sampler. Measurements were made, with an interval of 5 cm from the snow surface to the ground. In total, 13 samples were taken from the profile. The samples were weighed using an electronic scale (A&D HT-3000, readability 1 g).

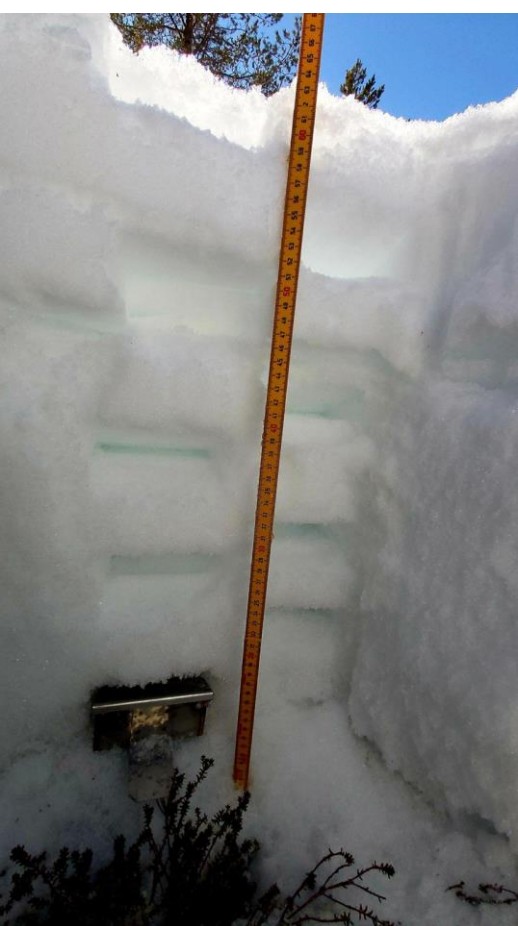

**Figure 3.** The density profile was measured using a 250 cm$^3$ RIP2 density cutter.

In addition, an automated SWE scale (Sommer Messtechnik SSG-2) recorded measurements at the same site, located approximately 5–10 m from the sampling area.

## 3. Results

The stratigraphy of the snowpack is presented in Figure 4, along with wetness, grain type, and hardness measurements for each layer. The snowpack was relatively soft and dry. The surface of the snowpack had a crust layer. Two moderately hard internal crust layers and an ice layer were found, which were due to the thawing periods that occurred during snow accumulation. Furthermore, the snowpack had a 15-cm layer of rounded grains, followed by an ice layer and a 33-cm layer of faceted crystals, while the bottom 10 cm comprised depth hoar.

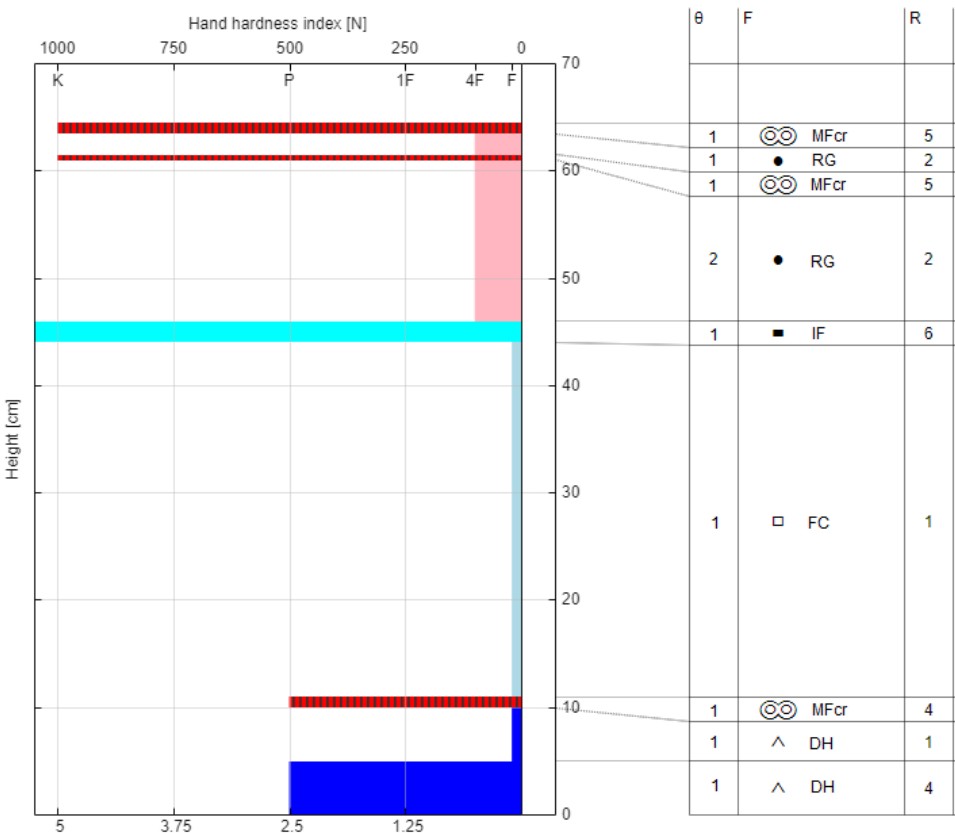

**Figure 4.** Stratigraphy of the snowpack. θ represents wetness, F represents grain type, and R represents the hardness of the snow, according to a classification in [25]. This visualization was created using the niViz tool.

The samplers and their measurement results are listed in Table 1. The snowpack density, based on the tube sample measurements, was in the range of 245 to 265 kgm$^{-3}$ (with an outlier for the University of Iceland Federal Sampler of 122 kgm$^{-3}$). The density profiles, based on three 5–100-centimeter resolution cutters, were from 240 to 320 kgm$^{-2}$. The measurement results of SWE were considerably different, depending on the sampling equipment used: they ranged from 77 to 180 kgm$^{-2}$, i.e., they were different by a factor of 2.3. The snow density cutter with a volume of 500 cm$^3$ had the largest SWE value of all the samplers and cutters. The RIP1 cutter produced value closer to the tube SWE samplers that were compared in this study. The values within each particular tube sampler's series were fairly stable: standard deviations were in the range of 2–15 kgm$^{-2}$; in one exceptional case, the value was 26 kgm$^{-2}$ (Table 1). The SWE value appeared fairly stable in those cross-sections that were of 40 cm$^2$ or larger, but they were considerably smaller for the smallest cross-section of 11 cm$^2$ (Figure 5). For those cross-sections of 38–100 cm$^2$, the SWE value was fairly stable; however, the Federal Samplers (with a cross-section of only

11 cm$^2$) indicated considerably smaller SWE values of 106 and 77 kgm$^{-2}$. Their standard deviations are rather high, suggesting a random error during the sampling process. The third-smallest SWE (118 kgm$^{-2}$) was measured using a RIP2 cutter; however, this was for an obvious reason since the lowest 10 cm of the snow package was omitted due to the presence of vegetation.

**Table 1.** Parameters of the samplers used and the measured values of SWE and snow density.

| Institution | Sampler | Area, cm$^2$ | Number of Cores/Profiles | SWE, kgm$^{-2}$ | | Density, kgm$^{-3}$ | |
|---|---|---|---|---|---|---|---|
| | | | | Average | St. Dev. | Average | St. Dev. |
| Tube samplers | | | | | | | |
| Uni. Tartu | deposition sampler | 44 | 3 | 151 | 2 | 253 | 9 |
| FMI | FMI Korhonen–Melander | 100 | 8 | 160 | 3 | 260 | 6 |
| Arctic Centre | AC white | 79 | 8 | 152 | 10 | 237 | 17 |
| Arctic Centre | AC transparent | 69 | 8 | 157 | 7 | 246 | 8 |
| ELY-centre | ELY Korhonen–Melander | 100 | 10 | 148 | 9 | 245 | 8 |
| Syke | Syke Korhonen–Melander | 100 | 8 | 159 | 15 | 245 | 8 |
| Uni. Iceland | Federal Sampler Iceland | 11 | 8 | 77 | 12 | 122 | 19 |
| Greenland Survey | Federal Sampler Greenland | 11 | 10 | 106 | 26 | 252 | 11 |
| Density cutters | | | | | | | |
| Greenland Survey | RIP1 cutter | 200 | 1 | 159 | - | 265 | - |
| FMI | 500 cm$^3$ cutter | 100 | 1 | 180 | - | 277 | - |
| FMI | RIP2 cutter | 100 | 1 | 118 | - | 268 | - |

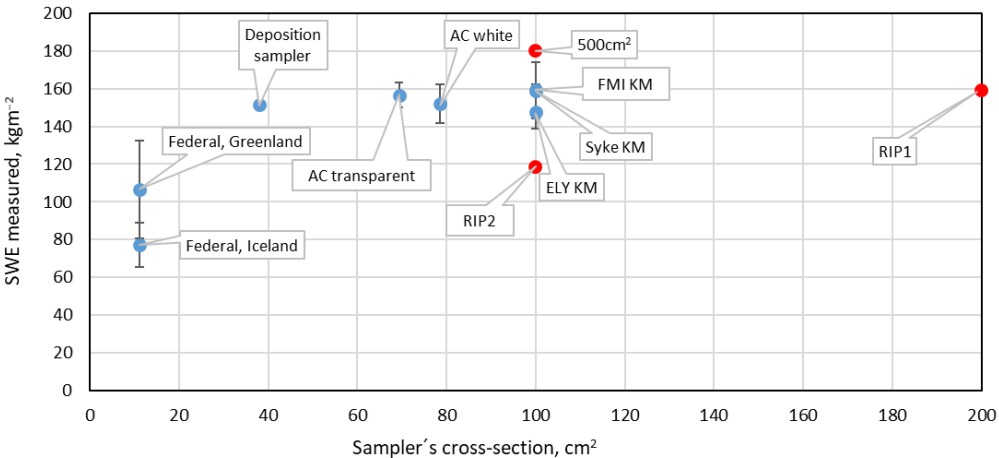

**Figure 5.** Dependence of the measured snow water equivalent on the sampler's cross-section, measured using tube samplers (in blue) and density cutters (in red). The error bars indicate the standard deviations of the series, consisting of multiple samples taken using the same sampler.

The automated SWE scale recorded an SWE of 157 kgm$^{-2}$, on average, for the same time interval as that when the manual measurements were made. This value is close to those in the results of the SWE samplers, excluding those of the Federal Samplers (148–160 kgm$^{-2}$).

## 4. Discussion

Our results present a relatively good comparison of the SWE measured between samplers, using cross-sections of >30cm$^2$ from the sampled snowpack. For these samples, the average SWE varies by 8%. It is encouraging to see that the layer-wise density methods and the tube values agree so well. The average SWE, based on the samples from two Federal Samplers, is 42% lower than the average of 100 cm$^2$ (the largest tube samplers).

Various types of sampling tubes may modify the physical properties of the snow found locally in the volume of sample caught within and immediately next to the outer surface of the sampler: in passing through the snow, the sampler tube compresses the snow, both inside and outside the tube. However, outside the tube there is plenty of space, resulting in less compression than that found inside the tubes. Due to this unequal compression rate, the drag force inside the tubes appears larger than that outside, partially preventing the snow that is under the cross-section of the sampler from being captured. In other words, the snow inside the sampler becomes partially clogged and a fraction of the snow in the path of the sampler is pushed aside. Obviously, the drag force inside the tube is greater if the diameter of the tube is smaller, relative to its wall thickness. This is the case with the Federal Sampler, a massive metal tube that is only 3.8 cm in diameter, which is designed to cut through a thick and icy snowpack. The device is estimated to have an error rate of 5–11%, depending on the snow conditions [26]. Previous field comparisons showed that the Federal Sampler yielded values that were 57% lower than the site's mean SWE, as measured by other devices [14]. This difference in the results between the two Federal Samplers and the density profile cutters compares very well with the differences reported by Leppänen et al. [27].

In this study, however, there are fairly substantial variations in the results of both tube samplers and layer-wise density cutters (see Figure 5). Some variations have been observed in the results of both tube samplers and layer-wise density cutters in this study (see Figure 5). The study indicates that the differences in the SWE values measured by different groups, even when using similar samplers, can be rather large. Variations were approximately ~30% for the Federal Sampler, ~9% for the other tube samplers, and ~13% for the density samplers. These variations can be attributed to several factors such as the measuring technique, including the speed and rotation of the sampler during its insertion into the snowpack. Due to the low density of the measured snowpack, even small changes in the snowpack may cause large relative errors in the SWE.

Our results show values similar to those recorded for instrument intercomparison by Lopez-Moreno et al. [20], wherein the Federal Samplers underestimated the snow density and SWE values, compared to other instruments on specific flat areas. Conversely, other studies have shown that in forests and areas with greater snow depth, Federal Samplers seem to be in good agreement with other snow tubes and may even give the highest snow density and SWE values [20,26,27].

## 5. Conclusions

For the snowpack that was measured in this study, the results for tube samplers with a cross-section of 40 cm$^2$ and greater (with a measured SWE of 148–160 kgm$^{-2}$) are comparable to those from sampling with density profile cutters (with a measured SWE of 159 and 180 kgm$^{-2}$). However, we found that the thinner sampler tubes, such as the two Federal Samplers of 11 cm$^2$, yielded only values of 77 and 106 kgm$^{-2}$. Thus, we report that thin samplers will underestimate the SWE value in certain conditions, such as a loose snowpack. The physical reason for this underestimation may lie in the interaction of the tube with the sampled snow, thereby changing the mechanical properties of the snow within and immediately next to the sampling tube when taking the samples.

Our study shows that the accuracy of sampling depends on the sampler diameter and possibly the thickness and material of the sampling cylinder, as well as the technique used to take the samples. Further comparative studies are needed to quantify the impact of these variables on the sampling accuracy. The measurements could be repeated using the same instruments under different conditions, e.g., in alpine areas or on a glacier, and the researchers could then make recommendations as to which device is best suited to particular circumstances. In our test location at Sodankylä, which is north of the Arctic Circle and is part of the boreal forest zone, we experienced difficult sampling conditions due to the fragile depth hoar layer and the vegetation found in the deepest layers, 0–10 cm from the bottom. This made sampling challenging, especially in the case of snow sampling

using density cutters and thin sampling tubes. Potentially, it would be possible to develop correction formulas for the these sampling tubes. More research under different snow conditions is needed to create a relevant dataset.

**Author Contributions:** Conceptualization, O.M.; methodology, O.M.; writing—original draft preparation, M.K., writing—review and editing, M.K., O.M., P.D.-W., K.L., L.L. and A.N.A.; software, M.K., validation, M.K.; formal analysis, M.K.; investigation, M.K., K.A., P.D.-W., T.H., L.L., Y.L., A.G. (Andri Gunnarsson), K.L. and O.M.; resources, M.K., K.A., P.D.-W., A.G. (Anders Ginnerup), T.H., L.L., Y.L., A.G. (Andri Gunnarsson), K.L. and O.M.; data curation, M.K.; visualization, M.K. and O.M.; supervision, M.K., O.M. and A.N.A.; project administration, O.M. and A.N.A.; funding acquisition, A.N.A. All authors have read and agreed to the published version of the manuscript.

**Funding:** This study is funded by Sustainable Development in the Arctic—The Nordic Council of Ministers' Arctic Cooperation Programme 2018–2021, project number A19200. This work was also partly funded by Orkurannsóknasjóður (National Power Agency of Iceland), the Estonian Research Council, grant PRG1726, and the European Commission, Horizon 2020 Framework Programme (CHARTER project grant no. 869471).

**Data Availability Statement:** The research data are available in Appendix A.

**Acknowledgments:** We would like to thank the Finnish Meteorological Institute personnel in Sodankylä for their invaluable support during the NordSnowNet Snow Week.

**Conflicts of Interest:** The authors declare no conflict of interest. The funders had no role in the design of the study; in the collection, analyses, or interpretation of data; in the writing of the manuscript; or in the decision to publish the results.

## Appendix A. List of Individual Snow Samples and Summary Table

https://nordsnownet.fmi.fi/documents/ (accessed on 2 May 2023).

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
