# Peer review of "Accuracy of Manual Snow Sampling, Depending on the Sampler’s Cross-Section—A Comparative Study"

_geosciences, doi:10.3390/geosciences13070205_

Round 1

Reviewer 1 Report

The manuscript by Kaasik et al. compare the snow measurements accuracy obtained by various traditional manual instruments used in the Nordic region (here Estonia, Finland, Iceland, Sweden, Greenland) based on a short field campaign in the Sodankylä area in northern Finland. The idea itself is not novel, however there are few scientific articles comparing snow water equivalent (SWE) measurements obtained from different class of equipment: SWE tubes and density cutters, what was done here by the authors. The work is similar to study by Lopez-Moreno et al. 2020, however introduce comparison of instruments that were not tested in this previous study.

Although idea of the study is good and results can be interesting for the scientific community, the manuscript is sloppy written and needs a lot of revision. There is a confusing section in the introduction about the chemical analysis of snow, which also introduces redundant citations for the main topic. In general, I have the impression that some of the citations in the work are used incorrectly. The manuscript itself is pretty short and uses data obtained during just one field day. Comparison of values obtained from SWE tubes and density cutters are interesting, however it is really hard to get meaningful statistics from such a scarce dataset. In result, the Authors conclude that tube samplers with small cross-section underestimate SWE, but this is effect of snow conditions (quite shallow and loose snowpack) in the study area. It would be great to repeat measurements with the same instruments in different conditions, e.g. Alpine or on the glacier, and then made recommendations on which device is best suited to the circumstances. In present form the manuscript is more a short project report than original research article.

More detailed comments on the authors' work can be found in the attached pdf file.

Author Response

Although idea of the study is good and results can be interesting for the scientific community, the manuscript is sloppy written and needs a lot of revision. There is a confusing section in the introduction about the chemical analysis of snow, which also introduces redundant citations for the main topic.

The section on chemical analysis is shortened and moved to the end of Introduction.

In general, I have the impression that some of the citations in the work are used incorrectly.

Citations are corrected.

The manuscript itself is pretty short and uses data obtained during just one field day. Comparison of values obtained from SWE tubes and density cutters are interesting, however it is really hard to get meaningful statistics from such a scarce dataset. In result, the Authors conclude that tube samplers with small cross-section underestimate SWE, but this is effect of snow conditions (quite shallow and loose snowpack) in the study area.

Several samples were taken with each tube sampler for statistical evaluation – variability is found rather small. The value of this study is in comparison of several samplers, which are in use in different countries, during single day in single site. The site is very typical for Northern boreal forest in flat landscape. This ecosystem extends over large areas in Eurasia and North America. Explanation is added to the text, which is extended now.

It would be great to repeat measurements with the same instruments in different conditions, e.g. Alpine or on the glacier, and then made recommendations on which device is best suited to the circumstances. In present form the manuscript is more a short project report than original research article.

Certainly, that would be worth to be done! However, the boreal forest, conditioning rather uniform, soft and low-density snow, is an area contributing through its albedo and cooling effect a lot to the global weather and climate system. The introduction is extended, keeping in mind these issues. We consider it worth to publish these results now, making them available for worldwide community, but indicating clearly the limitations of our dataset.

Specific questions/remarks are answered in comments in submitted revised manuscript.

Reviewer 2 Report

A comparative study of 12 different snow sampling methods was carried out in field. This is relatively simple but meaningful experiment. 

There are some small errors like double space, two dots at the end of a sentences. Please rectify these small errors. 

Basically there are two large types of samplers. For more easily read, I suggest the authors revise table 1 to indicate two types samplers: 9 bulk tube samplers, 3 density cutters. 

The density cutter results and tube samplers result presented in Figure 5 could also differentiated with two different colors. 

Author Response

Basically there are two large types of samplers. For more easily read, I suggest the authors revise table 1 to indicate two types samplers: 9 bulk tube samplers, 3 density cutters.

The density cutter results and tube samplers result presented in Figure 5 could also differentiated with two different colors.

Both Table 1 and Figure 5 are revised to distinguish the sampler types.

Reviewer 3 Report

Accurate in-situ observations of SWE are important for satellite-derived snow products, snow melt runoff simulations, climate change, etc. The results of this study are helpful to improve the accuracy of SWE observation. However, I think this manuscript is not complete enough, and some experiments and analysis need to be added. The authors have conducted only one comparative trial (one day and one site) so far. I suggest that the authors should increase the number of experimental tests and the number of sites to reduce the random error in the observation process. The authors should compare the influence of snow with different nature on the sampling accuracy of samplers. For example, whether new snow or repeated melting and freezing snow will affect the accuracy of samplers, to what extent, and whether it will affect the results of this manuscript.

Author Response

However, I think this manuscript is not complete enough, and some experiments and analysis need to be added. The authors have conducted only one comparative trial (one day and one site) so far. I suggest that the authors should increase the number of experimental tests and the number of sites to reduce the random error in the observation process.

Statistical variability due to sampling inaccuracy and local inhomogeneity is quantified by means of standard deviations based on multiple samples and found relatively small, see Figure 5.

The authors should compare the influence of snow with different nature on the sampling accuracy of samplers. For example, whether new snow or repeated melting and freezing snow will affect the accuracy of samplers, to what extent, and whether it will affect the results of this manuscript.

The samples were taken from snow cover accumulated through nearly 5 months, using the advantage of stable frosty winter in sub-arctic region. Thus, the results are expected representative for site type, not depending a lot on short-term events and weather extremes. Indeed, repeating the study through several years would add value of interannual variability to the intercomparsion results. However, we consider the existing results worth to publish soon, not with delay of several years, as there are not many data on accuracy of different in-situ samplers and sampling methods in northern boreal forest.

Round 2

Reviewer 3 Report

The authors have replied to my modification suggestions. I suggest accepting the manuscript.